# Unveiling Cancer Metabolism through Spontaneous and Coherent Raman Spectroscopy and Stable Isotope Probing

**DOI:** 10.3390/cancers13071718

**Published:** 2021-04-05

**Authors:** Jiabao Xu, Tong Yu, Christos E. Zois, Ji-Xin Cheng, Yuguo Tang, Adrian L. Harris, Wei E. Huang

**Affiliations:** 1Department of Engineering Science, University of Oxford, Oxford OX1 3PJ, UK; tong.yu@spc.ox.ac.uk; 2Molecular Oncology Laboratories, Department of Oncology, Weatherall Institute of Molecular Medicine, John Radcliffe Hospital, Oxford University, Oxford OX3 9DS, UK; christos.zois@oncology.ox.ac.uk; 3Department of Radiotherapy and Oncology, School of Health, Democritus University of Thrace, 68100 Alexandroupolis, Greece; 4Department of Biomedical Engineering, Boston University, Boston, MS 02215, USA; jxcheng@bu.edu; 5Suzhou Institute of Biomedical Engineering and Technology, Chinese Academy of Sciences, Suzhou 215163, China; tangyg@sibet.ac.cn

**Keywords:** cancer metabolism, Raman spectroscopy, stimulated Raman scattering, coherent Raman anti-Stokes scattering, Raman imaging, lipid metabolism, stable isotope probing

## Abstract

**Simple Summary:**

Raman spectroscopy and imaging are label-free, non-destructive techniques to study cellular metabolism with subcellular spatial resolution. This review focuses on applications of Raman-based methods in a combination of stable isotope probing on cancer metabolism and cancer imaging.

**Abstract:**

Metabolic reprogramming is a common hallmark in cancer. The high complexity and heterogeneity in cancer render it challenging for scientists to study cancer metabolism. Despite the recent advances in single-cell metabolomics based on mass spectrometry, the analysis of metabolites is still a destructive process, thus limiting in vivo investigations. Being label-free and nonperturbative, Raman spectroscopy offers intrinsic information for elucidating active biochemical processes at subcellular level. This review summarizes recent applications of Raman-based techniques, including spontaneous Raman spectroscopy and imaging, coherent Raman imaging, and Raman-stable isotope probing, in contribution to the molecular understanding of the complex biological processes in the disease. In addition, this review discusses possible future directions of Raman-based technologies in cancer research.

## 1. Introduction

Reprogrammed metabolism is considered a hallmark of cancer [1]. Cancer metabolism is a question of great interest in a wide range of fields since the “Warburg Effect” indicated aerobic glycolysis, as a characteristic of cancer cells [2]. Cancer cells rely on the acquisition of nutrients from the environment to meet the demand for energy and biomass production. In the last few decades, features of tumor-associated metabolic alteration were widely observed. Key questions among the tumor metabolic reprogramming rapidly expanded our understanding of carcinogenesis. In particular, how cancer cells reprogram their metabolism and interact with other biological processes, which mechanisms and functions of nutrient acquisition in cancer cells are achieved, and what targeted pathways in metabolism can be engaged to selective development of inhibitors during therapeutic evaluation [3].

Cancer metabolism is highly complex and heterogeneous, subject to environmental cues. The metabolic phenotype is dependent on the tumor microenvironment. It is now clear that even though cancer cells are from the same tumor with the same genotypes, they can manifest distinct phenotypic states in different positions [4]. Challenges remain in investigating the vastly complex metabolism of cancer with regards to intra-tumoral heterogeneity, different tissues, and tumor types.

Metabolomics based on mass spectrometry is a gold standard for study of cellular metabolites. Although lagging behind other single-cell omics methods, it recently advances to the level where multiplex analysis from single cells could be achieved [5,6]. However, the disadvantages of mass spectrometry metabolomics remain due to its destructive nature, prohibiting live cell analysis. Moreover, it does not offer insights on the spatial information of metabolite distribution, which can be crucial in heterogeneous intra-tumor environments. Fluorescence-based methods offer high sensitivity and selectivity, and they can provide subcellular resolution. However, the fluorophore labels are often much larger than the targeted molecules and thus can hinder the intrinsic metabolism [7]. The prerequisite of knowing the target molecules also hinders multiplex targeting and investigative probing. Considering the high complexity and heterogeneity in cancer, minimally invasive, non-destructive, and label-free tools with imaging ability providing subcellular spatial information, are in need to study cancer metabolism and fulfill the translational clinical need.

Raman spectroscopy technologies present several advantages for the cancer research community. In particular, Raman spectroscopy is a label-free spectroscopic method to provide comprehensive biochemical information, the so-called “biomolecular fingerprint”, regarding metabolic processes in cells and tissues. In addition, Raman imaging offers high spatial resolution at the sub-cellular level and is capable of real-time, noninvasive examination of living cells, tissues, and organisms, based on their metabolic profiles.

This review aims to present Raman-based technologies in the area of cancer research, to study the metabolism of single cells as well as physio-pathological tissues. First, we present applications of label-free Raman spectroscopy and imaging in cancer metabolism. Second, we discuss Raman-based technologies combined with stable isotope probing (SIP) for studying metabolic fates of specific metabolites. The primary focus of the methodology is on spontaneous Raman spectroscopy and coherent Raman scattering (CRS) microscopy. Readers interested in surface-enhanced Raman scattering and other Raman-based methodologies are referred to excellent reviews elsewhere [8,9,10,11].

## 2. Background of Raman Spectroscopy and Coherent Raman Scattering

### 2.1. Spontaneous Raman Spectroscopy and a Cell’s Fingerprint

Raman spectroscopy is an essential tool for chemists, physicists, biologists, and materials scientists. It was first experimentally observed by C. V. Raman in 1928, for which he was awarded the Nobel Prize in 1930 [12]. In Raman spectroscopy, a sample is illuminated with a monochromatic laser beam (Figure 1A). The incident laser beam with a frequency of υ*_0_* interacts with a molecule in the sample that has a vibrational frequency of υ*_m_*. This interaction originates scattered light. The vast majority of the scattered light is elastic scattering, the so-called Rayleigh scattering. However, due to energy exchange between the incident photons and molecular vibration, there is a small fraction of inelastic scattering, which has a different frequency from that of the incident light, collectively called Raman scattering. Raman scattering consists of both Stokes scattering, which has a lower frequency at υ*_0_ −*
υ*_m_*, and anti-Stokes scattering, which has a higher frequency at υ*_0_ +*
υ*_m_* (Figure 1B) The Stokes spectrum is often presented as the Raman spectrum, due to its higher intensity, as compared to the anti-Stokes lines.

Raman spectroscopy is now increasingly popular among biologists. A Raman spectrum can be regarded as a phenotype of a biological system because it provides an overall molecular vibrational profile, containing Raman bands for major cellular building blocks, such as proteins, nucleic acids, lipids, and carbohydrates. Figure 2 illustrates a Raman spectrum of a single glioblastoma cell with Raman bands marked with assignments of major biological macromolecules. A biological Raman spectrum can be divided into three regions—the ‘fingerprint’ region that contains essential bio-information and can be seen as a fingerprint of a cell (400–1800 cm^–1^); the ‘silent’ region that usually does not involve vibrational modes contributed by biomolecules formed of naturally occurring isotopes and can involve bands contributed by stable isotopes or triple bonds (1800–2700 cm^–1^); the high-wavenumber region that is specifically contributed by the stretching vibrations of CH groups, predominantly from lipids and proteins (2700–3200 cm^–1^).

The advantages of using Raman micro-spectroscopy in biological studies are high spatial resolution; ability to detect aqueous samples; intrinsic and label-free characterization; non-contacting and non-destructive analysis; and easy preparation and small sample volume.

By combining the power of optical magnification and direct visualization, Raman micro-spectroscopy can probe biological systems at a subcellular resolution. For large biological systems like tissues, it can collectively produce label-free Raman images with subcellular structural and chemical information. The ability to analyze aqueous samples separates Raman spectroscopy from other vibrational spectroscopies like infrared (IR) spectroscopy. A water molecule has low polarizability, thus, it minimally hinders sample signals in Raman spectroscopy and can be easily subtracted during preprocessing procedures. This is particularly advantageous in biological studies because it avoids laborious sample drying preparation, which might also alter the biochemistry of the samples.

Molecular labeling using genetically encoded reporters (e.g., green fluorescent protein) is used extensively for monitoring cellular events in biological systems [13]. The introduction of unnatural functional groups might risk interfering with the native biological processes. Raman spectroscopy, on the other hand, is label-free and able to probe biological samples in their natural setting. It also does not require any prior knowledge of the particular substrate for selective labeling. It unbiasedly probes all macromolecules and collectively displays their vibrational modes in one spectrum. Raman spectroscopy is also non-contacting and non-destructive, as opposed to other analytical tools such as gas chromatography and mass spectrometry that destroy the sample to achieve results. This non-invasive, label-free, and spatially resolved nature renders Raman spectroscopy suitable for in vitro and in vivo biological investigations, from investigating a single cancer cell, to producing biochemical maps of cancerous tissues.

### 2.2. Coherent Raman Scattering for High-Speed Imaging

Having discussed the advantages of Raman micro-spectroscopy, the challenge is intrinsically weak Raman scattering as approximately only 1 in 10^7^ photons experiences inelastic scattering [14]. Another Raman-based approach is emerging in recent decades to obtain much stronger vibrational signals by coherent Raman scattering (CRS), which employs multiple light sources to produce coherent Raman signals. 

Coherent anti-Stokes Raman scattering (CARS) and stimulated Raman scattering (SRS) are two CRS processes [15,16,17,18]. During spontaneous Raman scattering, the pump beam with a frequency of *ν*_0_ is incident upon the sample generating a Stokes signal *ν_S_* and an anti-Stokes signal *ν_aS_*. In SRS, two laser beams at frequencies *ν_0_* and *ν_S_* are directed onto the sample, such that the frequency difference *ν_0_ − ν_S_* matches the frequency of a molecular vibrational mode *ν_m_* (Figure 1B). This process causes stimulated excitation of a chemically specific signature and brings the advantage of suppressing non-resonant background. Experimentally, either stimulated-Raman gain (SRG) of the Stokes or stimulated-Raman loss (SRL) of the pump is measured, and optical modulation and demodulation are required to extract the proper SRS signal [19].

CARS involves a complex four-beam mixing process probing, as shown in Figure 1B. As in SRS, the resonance occurs when the difference between the pump and Stokes beam *ν_0_ ∓ ν_S_* matches a molecular vibration in the sample. This resonance is then probed by the third field at *ν_0_* and generates an anti-Stokes signal at *2ν_0_ ∓ ν_S_*. With respect to SRS, no complex demodulation schemes are required in CARS. However, the presence of a strong non-resonant background due to the four-wave mixing that does not carry any chemical information can distort and overwhelm the resonant signal [20].

With the advantages of dramatically enhanced Raman signal and little autofluorescence interference, CARS and SRS microscopy are emerging techniques for real-time analysis and video-rate imaging of cells and tissues in a living system, with high speed and 3D spatial resolution [21]. Nonetheless, both techniques are limited to the complex set-up of two synchronized trains of laser pulses. CARS and SRS are also limited by the small spectral range due to the narrow bandwidths between lasers. Therefore, so far in most cases, these were demonstrated for chemical signatures with high concentrations and strong Raman signals, for example, the CH_2_ stretching vibrations that are typically found in lipids in cells [22]. In addition, CARS suffers from the complicated interpretation of Raman bands due to changed band frequencies [20]. SRS, on the other hand, shows a comparable spectrum like spontaneous Raman. Its signal also shows an intensity that is linearly proportional to the concentration of the targeted molecules and is free of non-resonant background. With the intensity of the signal proportional to the concentration of the metabolites of interest, semi-quantitative molecular profiling of single cells and tissues is possible [23,24,25,26,27]. SRS would be a complementary tool to spontaneous Raman scattering and simultaneously using the two can be beneficial (Table 1) [28]. The investigation of a biological system can begin with spontaneous Raman scattering (acquisition time from seconds to minutes) covering the entire spectral window of molecular vibrations and looking for a defined marker of interest. It could then be followed with SRS focusing on a predefined spectral window and offering high-speed benefit (acquisition time microseconds) for imaging and collection of a huge amount of spectral data. 

## 3. Raman Spectroscopy as a Label-Free Tool for Cancer Metabolism Investigation

Metabolic reprogramming is now a widely accepted hallmark of cancer [3]. Monitoring metabolites and understanding the reprogramming pathways are crucial to fully understand cellular metabolism in cancer development and to provide new therapeutic targets. Raman spectroscopy and Raman imaging benefit cancer research and understanding of metabolic regulation in cancers, because it offers single-cell resolution or spatial information about biochemical composition of nucleic acids, proteins, lipids, and other metabolites. The defined subcellular locations cannot be provided by conventional analytical methods that rely on bulk or fractionated analyses of extracted components, e.g., mass spectrometry, nuclear magnetic resonance spectroscopy, and high-performance liquid chromatography. As a label-free technique, Raman spectroscopy also outperforms many fluorescence-based approaches that involve exogenous probes and can interfere with intrinsic biological processes.

### 3.1. Investigation of Lipid Metabolism in Cancer Cells

Lipid reprogramming is one of the key features of cancer’s metabolic adaptation [29]. Alterations in lipid metabolism were observed in many different cancer types [30,31,32,33], and have important therapeutic inferences, as they affect the survival, dynamics, and response of cancer cells. Raman microscopy is one of the most powerful techniques for analyzing the properties of lipids in cancer cells, offering lipid compositional information as well as spatial information within cellular compartments. As the lipidic CH stretching vibrations are the strongest among all Raman vibrations generated in the high-frequency region of 2800–3200 cm^‒1^, lipid metabolism is possibly the most investigated subject by Raman spectroscopy.

To satisfy proliferation demand, cancer cells exhibit a higher content of cellular lipid content than normal cells. Lipids are particularly in demand as nutrient sources for energy supply, building blocks for membrane biogenesis, and as lipid-derived signaling molecules [34]. A number of studies found an altered lipid metabolism related to the progression of cancer by Raman spectroscopy [23,35,36,37,38]. Abramczyk et al. found an increasing content of lipid droplets in mildly malignant (MCF7) and malignant (MDA-MB-231) breast cancer cells, as compared to the non-malignant (MCF10A) cells [36]. They also demonstrated altered lipid metabolism in adipocytes of the breast tissue from breast epithelial cells. Nieva et al. used Raman micro-spectroscopy to characterize lipid metabolism of breast cancer cells with different degrees of malignancy [39]. By analyzing characteristic Raman bands related to lipid content at 3014, 2935, 2890, and 2845 cm^−1^, they hypothesized that the lipid content of breast cancer cells might be a useful indirect measure of a variety of functions coupled to breast cancer progression. A study by Tirinato et al. demonstrated the capability of functional characterization of lipid droplets in cancer stem cells (CSCs) by Raman spectroscopy, and confirmed lipid droplets as a distinctive mark of CSCs in colorectal cancer [40]. Similarly, lipid upregulation and reprogramming was observed by Raman spectroscopy in cells of prostate cancer [23], lung cancer [41], and melanoma [42,43].

Subcellular composition and distribution of lipids were the center of the subject by both CARS and SRS microscopy, due to strong contrast in the CH vibrations. The high-speed capacity of CRS realizes rapid biochemical–optical lipid imaging, which is comparable to the traditional optical methods. In 2003, Nan et al. first used CARS microscopy to image neutral lipid droplets (LDs) in live fibroblast cells [44]. In 2008, Freudiger et al. used SRS microscopy to visualize lipid distribution with depth information [45]. They monitored lipids along varying depths of mouse skin, as well as DMSO penetrating into the skin. Since then, CRS microscopy was used extensively in lipid imaging with structural diversity tightly associated with their biological functions [43,46,47,48,49,50,51,52,53,54,55].

Hyperspectral CRS microscopy was applied to study changes in lipid composition (e.g., saturated vs. unsaturated) of cancer cells and tissues, which plays an important role in cancer metabolism and development. For example, increased saturation in phospholipids markedly alters signal transduction, protects cancer cells from oxidative damage, and potentially inhibits chemotherapeutic drug uptake [56]. Wang et al. visualized a substantial amount of saturated lipids accumulated in liver cancer tissues, compared with the adjacent noncancerous tissues [49]. Li et al. reported significantly increased levels of unsaturated lipids in ovarian CSCs, as compared to non-CSCs (Figure 3) [48]. Subsequent experiments showed that inhibition of lipid desaturases effectively eliminated CSCs, suppressed sphere formation, and blocked tumor initiation capacity in vivo. With spontaneous Raman spectroscopy, followed by SRS microscopy and transcriptomics analysis, Du et al. investigated and imaged lipids droplets in patient-derived melanoma cells during differentiation [43]. They identified fatty acid synthesis pathway as a druggable susceptibility and a lipid mono-unsaturation within de-differentiated mesenchymal cells, with innate resistance to BRAF inhibition.

### 3.2. Investigation of Cellular Metabolism beyond Lipids

The applicability of Raman-based techniques is not limited to lipid in the high-frequency CH region. The fingerprint region between 300 and 1800 cm^‒1^ contains a collection of biological information of a cell. Investigative research focusing on the fingerprint region can either aim for a particular biomarker or use the whole region with chemometric techniques, to deconstruct Raman bands into biological information. Figure 4 shows an example of spontaneous Raman imaging of a HeLa cell, using the fingerprint region as well as the high-wavenumber region of Raman spectra.

The study of proteins can be conducted by investigating Raman bands of Amide vibrational modes, including Amide I that ranges from 1600–1670 cm^‒1^, Amide II that ranges from 1480–1580 cm^‒1^, and Amide III that ranges from 1230–1300 cm^‒1^. A study of two breast cancer and one normal breast cell lines (MDA-MB-436, MCF-7, and MCF-10A) illustrated decreased protein content in cancerous cell line [57]. A linear discriminant analysis (LDA) model on the entire spectral range predicted the three cell lines with 100% sensitivity and 91% specificity. Abramczyk et al. demonstrated a Raman biomarker of protein phosphorylation using the ratio between two Raman peaks at 1586 and 829 cm^‒1^, representing tyrosine phosphorylation [58]. They subsequently found overexpressed phosphorylation in the human breast, small intestine, and brain tissues, and in the glioblastoma U-87 MG cell line using this Raman biomarker. Using a multivariate curve resolution (MCR) to deconstruct the whole Raman spectra, Marro et al. demonstrated different intensities of proteins, lipids, and mitochondria Raman bands in primary breast cancer cell lines and their metastatic variants in bone [59]. Kopec et al. imaged and located glycogen, glycosaminoglycan, chondroitin sulfate, heparan sulfate proteoglycan, and distinguished each chemical species in normal and cancer tissues [60]. As a result, the study concluded that the metabolism of proteins, lipids, and glycans was markedly deregulated in breast (adenocarcinoma) and brain (medulloblastoma) tumors. Using a principal component–linear discriminant analysis (PC–LDA) to deconstruct the Raman spectra of primary normal breast cells, and immortalized, transformed, non-invasive, and invasive breast cancer cells, Chaturvedi et al. identified distinct clustering of cell types with a high degree of sensitivity [61]. A study by Lemoine included 547 in situ Raman spectra from 65 patients undergoing glioma resection and systematic literature analysis of Raman study of glioma [62]. They subsequently used band fitting for extraction of Raman features and identified oncogenic processes involved with increased nucleic acid content, overexpression of type IV collagen, and a shift in the primary metabolic engine.

Coherent Raman imaging of proteins besides lipids is also a common practice in both SRS and CARS microscopy. Imaging the CH moiety at the high-frequency region provides information of both proteins and lipids, as it is rich in both proteins and lipids. Therefore, protein images can be obtained indirectly by subtracting the lipid moiety contribution from CH-derived images [52,55,63,64,65]. In addition, protein molecules can be imaged at 1655 cm^–1^ at the Amide I peak [64]. The Xie group demonstrated that DNA distribution could also be retrieved from the strong background of proteins and lipids at the CH region through linear decomposition of the SRS images [65].

Molecular vibrations in the fingerprint region other than the Amide I group show markedly reduced intensities, compared to the high-frequency CH stretching region, which is less studied in CRS microscopy [55]. At the fingerprint region, Sunney Xie’s group demonstrated the first SRS imaging of DNA at the fingerprint region in living cells [64], by using Raman bands at 785 and 1090 cm^‒1^. Recently, advancement in instrumentation and data science enabled SRS metabolic imaging of cancer cells in the fingerprint region with enhanced sensitivity. Cheng’s group reported SRS imaging of retinoids with significantly boosted molecular sensitivity to 34 micromolar, via visible preresonance SRS (VP–SRS) microscopy [66]. By shifting the excitation laser wavelength to visible range, approaching the absorption of intrinsic chromophores, the study demonstrated heterogeneous distribution of retinoid storage inside various cancer cells and chemoresistant ovarian cancer cells. Combining advanced instrumentation using a polygon-based ultrafast delay scanner and advanced data science approach of a spatial–spectral residual learning network, the group further demonstrated 100 times improvement of signals in fingerprint SRS imaging. They illustrated distribution of proteins, cholesterols, and lipids, in live pancreatic cancer cells, as well as in a whole mouse brain [17].

### 3.3. Cellular Responses to Anti-Cancer Drugs and Radiotherapy

The uptake, metabolism, and distribution of a drug candidate in targeted cancer cells or tissues are pivotal during drug discovery and development. Raman spectroscopy and CRS microscopy were used to exploit metabolic transformation in cancer cells as a drug target. Jamieson et al. investigated responses of PC3 prostate cancer cells to a series of lipid-targeting drugs [67]. Compared to the non-cancerous cells, the beta-blocker propranolol selectively chose the cancer cells in their Raman lipid profiles, showing an unexpected anti-cancer potential. The cellular lipid content in response to the drug was also studied by CARS microscopy [68]. By imaging the subcellular lipid distribution in hormone-treated breast and prostate cancer cells, the researchers found an increased number and size of intracellular lipid droplets and a higher degree of saturation in treated T47D and LNCaP cells than untreated cells. 

In addition to lipids, comprehensive metabolic adaptations of cancer cells in response to anti-cancer drugs could be investigated by studying the entire Raman spectral region [69,70,71,72,73,74]. El-Mashtoly et al. applied Raman spectral imaging to study the effect of the epidermal growth factor receptor (EGFR) inhibitor panitumumab on cell lines expressing wild-type Kirsten-Ras (K-Ras) and oncogenic K-Ras mutations [70]. Larion et al. investigated the drug response of FK866, an inhibitor of NAD^+^ salvage pathway, with both a fibrosarcoma cell line and mouse models [71]. The anti-cancer drug doxorubicin was studied by Farhane et al. [72] and Zhang et al. [73] to characterize its interaction with cancer cells. Note that Raman-based techniques could be used in combination with other techniques to be capable of conducting rapid drug screening and discovery. Wen et al. combined results from Raman spectroscopy with mass spectrometry, which is the gold standard in metabolomics, and characterized breast cancer responders and nonresponders to small molecule inhibitors [69].

Intracellular drug distribution and metabolism are traditionally visualized by fluorescent-labeled drug molecules. Despite the molecular specificity, fluorescent labels are often much larger than the drug molecules themselves and could largely alter the drug pharmacokinetic activities. Raman imaging offers new possibilities for label-free drug imaging with subcellular spatial resolution. Aljakouch et al. reported the intracellular spatial distribution and metabolism of neratinib, a tyrosine kinase inhibitor with antitumor property, in different cancer cells, using label-free Raman imaging [75]. Using the intrinsic CN bond in neratinib, which generates a Raman band at 2208 cm^‒1^ in the silent region, the authors generated label-free images of neratinib distribution in cancer cells (Figure 5). A study by El-Mashtoly et al. used Raman microscopy to show the spatial distribution of the molecular-targeted drug erlotinib within the cell and that erlotinib was metabolized within cells to its demethylated derivative [76]. Recently, the advance of SRS microscopy enabled ultra-rapid images of cells and subcellular distribution of drugs. Fu et al. reported SRS visualization and quantification of two tyrosine kinase inhibitor drugs (imatinib and nilotinib), as well as the process of drug uptake into lysosomes [77]. Visualization of other drugs including ponatinib and retinol was also reported by SRS microscopy [15,78].

Besides chemotherapy, radiotherapy is the mainstay of the treatment for a range of types of cancer. Monitoring radiation-induced cellular response of cancer at both cell and tissue levels was studied by Raman spectroscopy [79,80,81,82,83]. Roman et al. reported a change in the lipid composition and concentration in prostate cancer cells after X-ray radiation [80]. Similarly, metabolic alterations of cancer cells after radiation were also seen in nasopharyngeal cancer [79], non-small lung cancer [83], and glioblastoma [82]. A study by Milligan et al. found different biochemical responses between radio-resistant and radio-sensitive cell types in lung (H460), breast (MCF7), and prostate (LNCaP) cells [81]. The main differences were found in glycogen, phosphatidylcholine, glucose, arginine, and asparagine based on restricted non-negative matrix factorization approach.

### 3.4. Potential Applications in Clinical Cancer Diagnosis

Noninvasive or minimally invasive in vivo tools that can provide rapid tissue assessment have potential application in clinical diagnosis of cancer. With many advantageous features of Raman spectroscopic methods (non-destructive, capable of deep penetration and high resolution, and chemical specificity), here, we discuss their potential in tissue imaging and potential clinical diagnosis. We also direct readers to a few more reviews on applications of Raman spectroscopy in cancer diagnosis [84,85,86].

A recent work by Contorno et al. compiled data from 41 papers aiming at characterizing breast cancer by using various modalities of Raman spectroscopy [87]. They identified aromatic amino acids as the most prominent biomarker for identifying cancerous breast tissues from their healthy counterparts. Cancer cells and tissues exhibited markedly higher expression of aromatic amino acids, specifically tryptophan, phenylalanine, and tyrosine. A study by Haka et al. acquired Raman spectra from ex vivo samples of human breast tissue (normal, fibrocystic change, fibroadenoma, and infiltrating carcinoma) from 58 patients [88]. The study reported an increase in collagen in all abnormal breast tissues and less fat content in the samples diagnosed as fibroadenoma than those diagnosed as infiltrating carcinoma. By using the different fit coefficients for fat and collagen, the authors demonstrated a Raman diagnostic algorithm illustrating 94% sensitivity, 96% specificity, and 86% overall accuracy for detecting infiltrating carcinoma. A range of other studies also demonstrated the potential of Raman spectroscopy in diagnosing breast cancer in human and mouse breast tissues [89,90,91,92,93,94,95,96]. By exploiting the fingerprint region of the Raman spectrum and various machine learning techniques, Raman spectroscopy showed diagnostic potential for distinguishing cancerous tissues from normal tissues in brain cancer [56,97,98,99,100], skin cancer [42,101,102,103,104,105], gastrointestinal cancer [106,107,108,109], and lung cancer [110,111].

The ability of CRS for generating rapid chemical images makes it ideal for tissue and whole organ imaging and subsequently localizing the tumor margins. Ji et al. reported the use of SRS imaging for differentiating healthy human and mouse brain tissues from tumor-infiltrated brain tissues [53]. They demonstrated that these label-free histopathological images provided very similar results to those obtained by conventional hematoxylin & eosin (H&E) staining, providing intrinsic chemical information without the need of routine tissue processing. The authors further applied SRS microscopy to obtain in vivo mouse brain images during tumor resection surgery, to reveal tumor margins that were undetectable under standard operative conditions. Through SRS microscopy, Freudiger et al. generated multi-color images of lipids, proteins, and hemoglobin in fresh-frozen tissue sections from mice models of invasive glioma, breast cancer metastases, stroke, and demyelination. A good correlation between SRS and H&E microscopy was also shown. These findings suggest that SRS microscopy can generate high-quality histological images for clinical diagnosis without the need for tissue fixation, sectioning, or staining. Noteworthy, with the recent advances of CARS and SRS endoscopy [112,113,114], further improvements to these Raman-based systems would open up exciting possibilities for in vivo, label-free, and non-invasive histopathological imaging and clinical diagnostics in the near future.

## 4. You Are What You Eat—Stable Isotope Probing (SIP)

The introduction of stable isotope probing to Raman spectroscopy was first demonstrated in bacteria [115], which was later referred as Raman–SIP technology [116]. Huang et al. showed that the incorporation of ‘heavy’ ^13^C stable isotope into cells causes significant shifts of some Raman bands in single cell Raman spectra (SCRS) [115]. Later it was found that other stable isotopes D (^2^H) and ^15^N also generated Raman shifts in SCRS at different positions [117]. Following those discoveries, Raman–SIP are found applicable to many biological models including cancer cells, revealing the metabolic activity of cancer.

Stable isotope labeling by amino acids in cell culture (SILAC) has become increasingly popular for accurate protein quantitation, by using isotopically labeled amino acids that are later metabolically incorporated into cells [118]. In cancer research, SILAC was also recognized as an important tool for metabolic profiling of cancer cells, with respect to their environments [119,120]. Techniques like SILAC are stable isotope probing (SIP) methods that involve exposing cells to isotopically labeled substrates that are consequently assimilated into cells that are actively involved in specific metabolic processes. Molecular analysis of the labeled biomarkers thereby reveals functional information about the cell responsible for the metabolism of a particular substrate.

SIP to study cell metabolism, in general, shows great advantages over other specific probes that have larger molecular weights, and can disturb the intrinsic metabolic activity of cells. Using isotopes that mimic their naturally abundant counterparts does not alter the natural substrate pool. Among all SIP techniques, Raman spectroscopy has its unique strengths. While modalities like isotope ratio mass spectrometry and NMR spectroscopy require at least 300,000 cells, Raman spectroscopy or secondary ion mass spectrometry (SIMS) coupled with SIP enables an analysis of cell functions at a single-cell level [121]. However, SIMS is limited by its destructive nature and expensive equipment. Raman spectroscopy exceeds other techniques in its non-destructive, sensitive, and relatively cheap nature. While a Raman spectrum already conveys intrinsic and phenotypic information, more insight into cell functions can be obtained by combining Raman micro-spectroscopy with SIP, using D (^2^H), ^13^C, and ^15^N-containing substrates to replace their primordial isotopes (^12^C, ^14^N, and ^1^H) [116]. Among those stable isotopes, D (^2^H) and ^13^C are mostly commonly used due to the stronger C–H bond strength. Heteroatomic X–H bonds such as O–H, N–H, and S–H can be exchanged from nonenzymatic reactions, while C–H bond formation depends solely on enzyme-catalyzed chemical reactions that occurred within the metabolic pathways [122]. This provides unique advantages of D (^2^H) and ^13^C for probing the biological metabolism. Providing a wide variety of possible isotopic substrates, Raman–SIP probe the metabolism of most biomolecular constituents of a cell, including proteins, lipids, carbohydrates, and nucleic acids, revealing both molecule-specific and general metabolisms, as well as drug–cancer interactions (Figure 6).

Although Raman–SIP is now a mature tool to study cell ecology and biology, its application in cancer research is new. Most applications demonstrate the capability of the technique rather than answer specific questions. One of this review’s purposes is to introduce Raman–SIP to a broader community and share our perspectives on its potential for studying cancer metabolism.

### 4.1. Principles of Raman–SIP

As Raman spectroscopy considers changes in vibrational modes of molecules, a mechanical model can be used to consider a molecular vibration as a spring model. A classical ‘two-balls-on-a-spring’ approach provides an equivalence of a diatomic molecule with two atoms with masses *m_1_* and *m_2_* connected by a chemical bond. The vibrational frequency υ of such a molecule is described as: $$\upsilon =\frac{1}{2\pi c}\sqrt{\frac{k}{\mu }}$$ where *c* is the speed of light, *k* is the force constant of the spring (diatomic bond), and *μ* is the reduced mass given by: $$\mu =\frac{m_{1}m_{2}}{m_{1}+m_{2}}$$. Therefore, the vibrational frequency υ is inversely proportional to the square root of the reduced mass *μ*. When an atom is replaced by its heavier isotope, *μ* increases, therefore, υ decreases. Taking C–D replacement of C–H as an example, if the ^1^H atom is replaced by the ^2^D atom, *μ* almost doubles and the ratio of the new υ to the old υ is 1.36. Therefore, C–D vibrations usually appear in the silent zone of a Raman spectrum at around 2100–2200 cm^−1^, red-shifted from the C–H vibrations at 2800–3000 cm^−1^. Thus, a strategy to quantify D incorporation is to calculate the percentage of the integrated C–D band area to the sum of the areas of the C–D and C–H bands: $$D_{\mathrm{incorp}}=\frac{{\mathrm{Area}}_{C–D}}{{\mathrm{Area}}_{C–D}{\;+\;\mathrm{Area}}_{C–H}}$$. Redshifts of Raman bands can also be observed when replacing ^12^C, ^14^N, ^18^O, or ^32^S with their heavier analogs. Compared with the C–D shifts, the extent of the redshift is considerably less dramatic in the ^13^C and ^15^N substitutions, due to the comparably low changes in μ. By feeding cells with isotope-labeled substrates, the active cells metabolize these substrates and the isotopic-dependent shifts in Raman spectra can be used as indicators of isotopic incorporation, thereby ‘you are what you eat’.

### 4.2. Raman–SIP Monitors Intracellular Metabolic Activities

Raman spectroscopy combined with isotopically labeled molecules is used to study the specific metabolic features of cell constituents such as lipids, proteins, and nucleic acids (Figure 6). Table 2 summarizes a collection of studies using Raman–SIP approaches to study cell metabolism and behaviors, together with their choices of probes, targeted molecules, and platform of research.

There is increasing evidence of upregulated demand for fatty acids in cancer cells, compared to their non-malignant counterparts [30]. Lipid uptake, distribution, and metabolism was extensively studied using Raman–SIP approach with labeled fatty acids in various types of cells [123,124,125,126,127,128,129]. D-labeled free fatty acids such as d_31_-palmitic acid, d_33_-oleic acid, and d_8_-palmitic acid were used to image lipid metabolism in human macrophages [123,124,125]. Intracellular cholesterol storage can also be observed in cells by using d_38_-cholesterol [128]. Heterogeneous distributions of neutral lipid species were found, where some lipid droplets accumulated preferentially unesterified cholesterol, whereas others stored cholesteryl esters.

De novo lipogenesis in cells can also be studied using deuterated carbon sources such as deuterated glucose or glutamine. Li and Cheng were the first to visualize the direct de novo lipid synthesis that originated from deuterated glucose through SRS imaging [127] (Figure 7A and 7B). They observed that glucose was largely utilized for lipid synthesis in pancreatic cancer cells, which occurred at a much lower rate in immortalized normal pancreatic epithelial cells. Similarly, de novo synthesis of fatty acids was also studied in undifferentiated and differentiated melanoma cell lines using deuterated glucose with spontaneous Raman spectroscopy and SRS [43]. In combination with transcriptomics analysis, the authors identified the fatty acid synthesis pathway as a druggable susceptibility for differentiated melanocytic cells.

Tumor-selective cytotoxicity of particular fatty acids was also studied. Dodo et al. synthesized several deuterated γ-linolenic acids and evaluated their metabolism and cytotoxicity towards normal human fibroblast WI-38 cells and VA-13 tumor cells [130]. Through Raman imaging of intracellular lipid droplets, they suggested the tumor-selective cytotoxicity of γ-linolenic acids from itself, as opposed to its oxidized metabolites.

Cancer cells exhibit significant metabolic alterations with respect to their primary energy sources, including glucose. Deuterated glucose in Raman–SIP are shown to be valuable for the evaluation and exploration of glucose metabolism in cancer cells. Zhang et al. generated a comprehensive study to visualize the metabolic dynamics of macromolecules, such as DNA, protein, lipids, and glycogen, from glucose, in various cancer cell lines, and a mouse model with glioblastoma xenograft [137] (Figure 7C). The pathway from glucose to glycogen, which is important glucose storage in cancer cells, was also investigated by subcellular visualization of deuterated glycogen from deuterated glucose [138]. Hu et al. developed a protocol for synthesizing deuterated glucose with an alkyne tag, 3-O-propargyl-D-glucose [139]. By using this tag that shows a distinctive Raman band at 2129 cm^‒1^, they quantified the glucose uptake and found different metabolic activities in different cancer cell types. Long et al. further developed this approach to add a second ^13^C labeling to synthesize ^13^C-3-O-propargyl-D-glucose [140]. Thus, they demonstrated two-color imaging of glucose uptake and incorporation activity in U-87 MG human glioblastoma cells, PC-3 human prostate cancer cells, COS-7 monkey kidney cells, and RWPE-1 human prostate normal cells, as well as in ex vivo mouse brain tissues.

Just as in SILAC, protein synthesis could be observed in Raman–SIP through the incorporation of isotopically labeled amino acids. Back in 2008, van Manen et al. demonstrated the incorporation of deuterated phenylalanine, tyrosine, and methionine into proteins in single HeLa cells, also observed by the C–D vibrations in the 2100–2300 cm^‒1^ spectral region [132]. It was shown that Raman images could be generated by illustrating newly synthesized proteins from deuterated amino acids, in a range of live mammalian cells with high spatial–temporal resolution [133]. Using this approach, subcellular compartments could be identified, revealing fast protein turnover in HeLa and HEK293T cells, and newly grown neurites in differentiating neuron-like N2A cells. Metabolic changes of proteins and lipids were also studied in cancer cells during epithelial–mesenchymal transition [142]. Compared with approaches like SILAC or fluorescence, this approach with Raman–SIP could examine the proteome of a cell in a spatially resolved, non-destructive manner on living single cells with only minor modifications [145]. This approach could also be used to study protein degradation using ^13^C labelled amino acids [134,135].

Compared with lipid and protein metabolism, there are fewer studies utilizing Raman–SIP approach to study DNA synthesis. Apart from newly synthesized DNA observed from deuterated glucose metabolism, EdU as an analog of thymidine was exclusively used as a DNA metabolic tag, which was incorporated into cellular biomass during cell division [146]. Chen et al. synthesized various designs of ^13^C isotopologues of EdU and demonstrated three-color chemical imaging of nascent DNA, RNA, and newly synthesized fatty-acid in live HeLa cells.

### 4.3. Raman–SIP with D_2_O Measures General Metabolic Activity

Apart from specifically labeled precursors of cellular constituents, heavy water (D_2_O) is suggested to be a unique and universal tool to monitor the synthesis of biomolecules on a global scale. As an isotopologue of water, D_2_O can rapidly and freely equilibrate with the total body water inside a cell, and D atoms can exchange with the H atoms to form a variety of X–D bonds. Different from the often non-enzymatic reactions to form heteroatomic X–D bonds, such as O–D, N–D, and S–D, the C–D bond formation depends solely on enzyme-catalyzed chemical reactions, due to the stronger C–H bond strength [122]. In C–H/C–D exchanges, D atoms from D_2_O are rapidly incorporated into metabolic precursors of different classes of biomolecules such as deoxyribose, acetyl-CoA/NADPH, amino acids, and phosphoenolpyruvate (PEP) (Figure 8). These precursors with D labeling are then slowly incorporated into their final products, which are nucleic acids, lipids, proteins, and carbohydrates. Therefore, D_2_O serves as an ideal agent to probe general metabolic activities through the emergence of a variety of D-labelled macromolecules.

Raman–SIP using D_2_O was demonstrated to measure the general metabolic activities of single microbial cells, first reported by Berry et al. [117]. By incubating microbes with D_2_O, physiologically active cells could be rapidly and sensitively identified with Raman, by measuring the C–D peak at 2170–2300 cm^−1^. The metabolic activity could then be semi-quantified and compared by measuring the degree of D incorporation. This approach was extensively applied to study microbial activity and ecology [116,121,147,148,149,150,151,152,153,154]. Recently, it also generated an impact in the area of metabolic studies of animal cells.

Shi et al. demonstrated SRS imaging based on D_2_O in various animal models, achieving dynamic visualization of proteins, lipids, and DNA [143]. Based on the fact that tumor cells show inherently higher metabolic activities than normal cells, the study was able to visualize the boundaries between tumors and the surrounding normal tissues. The distribution of intratumor heterogeneity was also observed, which is considered a driver of tumor aggressiveness. In this study, the authors also suggested that D_2_O is a better probe than deuterium-labeled carbon substrates. For visualizing lipogenic activities, D_2_O is better than deuterated fatty acids, as well as deuterated glucose, which at high concentrations, could potentially create hyperglycemia and might be less efficient in labeling newly synthesized lipids. In addition, D_2_O is cost-effective, as compared to other Raman–SIP probes and is universal in probing a range of metabolic activities simultaneously. Most recently, Hekmatara et al. applied Raman–SIP with D_2_O to study anticancer drug response of MCF-7 cells [144]. They discovered high subcellular activities of cancer cells after high-dosage rapamycin exposure at the single-cell level, which was masked in a population-wise cytotoxicity test.

Although Raman–SIP application in cancer research using heavy water is just coming of age, recent studies exhibit its great potential in metabolic studies, as it is non-invasive, cost-effective, easily accessible, and universally applicable. It can be adapted to a broad range of model systems and cancer research areas to study cancer development, tissue homeostasis, tumor heterogeneity, cancer drug response, and pharmacokinetics. It would be of particular value in diagnosis at surgery, categorizing cancers by their metabolic status, evaluating new drugs, their uptake and metabolism, their antitumor effects, resistance mechanisms, and responses. Characterizing the changes in stroma at the local and distant sites and in premalignant disease, is likely to provide new insights into cancer development.

## 5. Conclusions and Outlook

Raman spectroscopy technologies, from comprehensive spectroscopy to advanced imaging, offer exciting new possibilities in cancer research. In this review, we summarize a wide range of applications using Raman and Raman–SIP techniques to investigate metabolism in cancer cells and tissues. Offering intrinsic biochemical information and subcellular spatial resolution, the Raman profile of a cancer cell or tissue is an excellent metabolic indicator and an indicator of phenotypic heterogeneity. It can help uncover the molecular basis of the disease and provide comprehensive, objective, and quantifiable molecular information for diagnosis and treatment evaluation.

Looking into the future, the potential of Raman spectroscopy and imaging in cancer research could be further exploited. Critical improvements and advances are still under development. Raman databases of more metabolites, potentially comparable to a mass spectrometry database, can cover identification of more metabolites. Advanced chemometric algorithms and machine-learning methods can improve and accelerate current spectral processing and data interpretation, especially in a large, high-dimensional clinical dataset. Instrumentation engineering of advanced fiber optics, detectors, and handheld spectrometer would render Raman tools accessible for clinical translation.

## Figures and Tables

**Figure 1 cancers-13-01718-f001:**
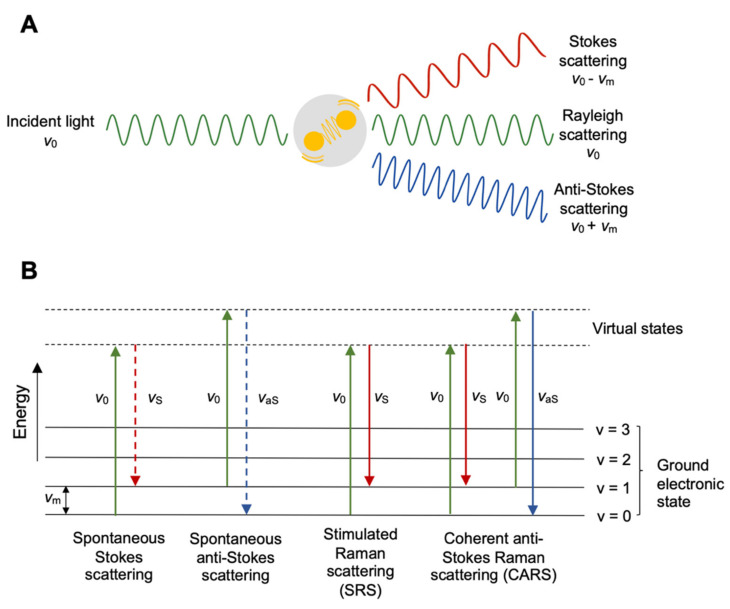
(**A**) Electromagnetic radiation interacting with a vibrating molecule. (**B**) Schematic energy diagrams of spontaneous Raman scattering and coherent Raman scattering (CRS). The solid arrows indicate laser excitation or stimulated emission; the dashed arrows indicate spontaneous scattering process.

**Figure 2 cancers-13-01718-f002:**
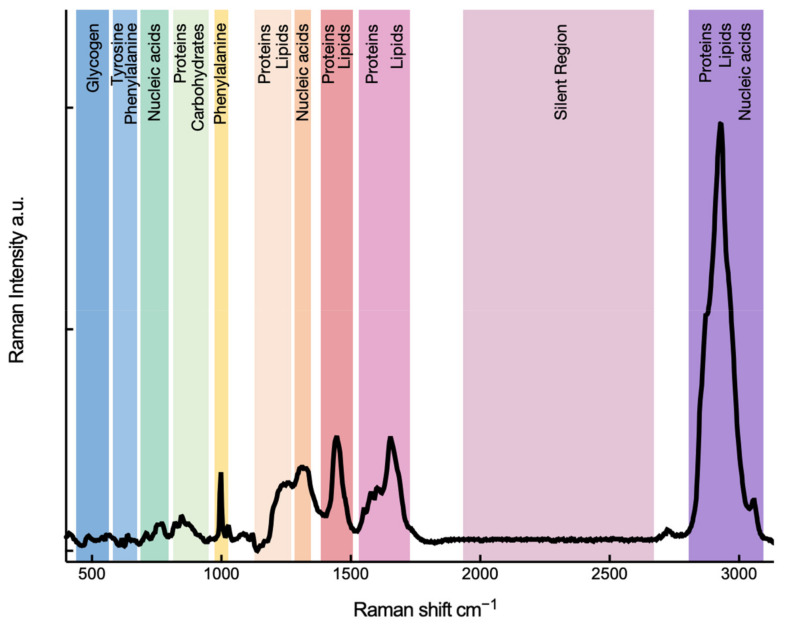
Raman spectrum—a cell’s fingerprint. Raman spectrum of a single cell of human primary glioblastoma U87 cell line, demonstrating various bands representative of cellular constituents.

**Figure 3 cancers-13-01718-f003:**
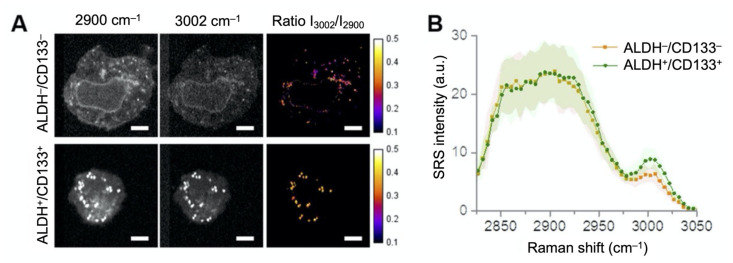
Increased lipid unsaturation level in ovarian cancer cells represented by intensity ratio between 2900 and 3002 cm^‒1^. (**A**) Representative hyperspectral SRS images of flow-sorted ALDH^−^/CD133^−^ and ALDH^+^/CD133^+^ COV362 cells. Images at 2900 and 3002 cm^‒1^, and the intensity ratio image between 3002 and 2900 cm^−1^ are shown. Scale bars: 10 μm. (**B**) Average SRS spectra from the lipid droplets in ALDH^−^/CD133^−^ (*n* = 3) and ALDH^+^/CD133^+^ cells (*n* = 8). Shaded area indicates the standard deviation of SRS spectral measurement from different cells. Reprinted and adapted with permission from reference [48].

**Figure 4 cancers-13-01718-f004:**
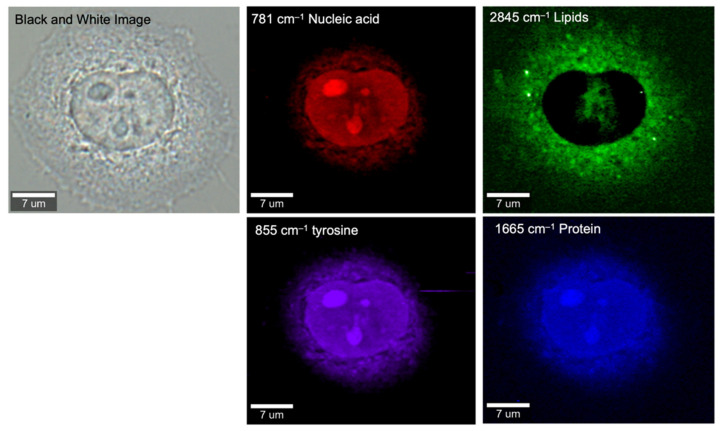
Raman images of a HeLa cell based on fingerprint Raman spectra, reconstructed from intensities at 781 (nucleic acids), 855 (tyrosine), 1665 (protein), and 2845 cm^−1^ (lipids). The Raman images show the distributions of these biomolecules with various intensities at the subcellular level.

**Figure 5 cancers-13-01718-f005:**
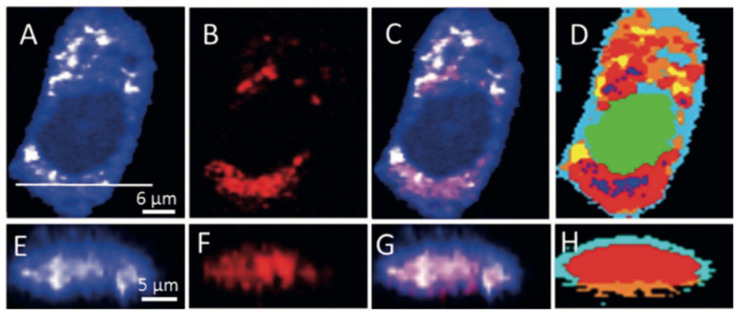
Raman imaging of SK-BR-3 cells treated with 5 µM neratinib for 8 h. Raman images reconstructed from the CH deformation (**A**) and CN stretching (**B**) intensities. (**C**) Overlay of Panels A and B. (**E**–**G**) Cross-section of Raman images of the same cell measured along the x–z axis. Scanning positions are indicated by the white line in Panel A. (**D**,**H**) HCA results based on the Raman data shown in Panels A and E. Reprinted with permission from reference [75].

**Figure 6 cancers-13-01718-f006:**
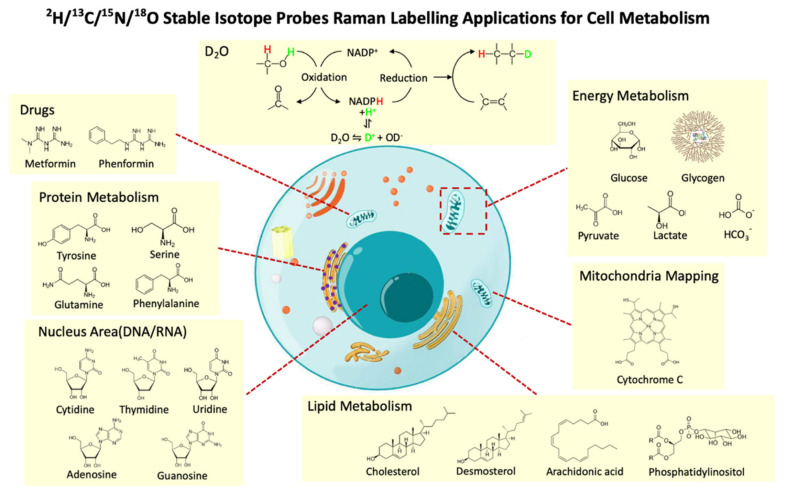
Raman–SIP strategies to study cellular metabolism. Heavy water (D_2_O) is involved in NADPH regeneration, which is able to indicate the general anabolism activity in cells. Stable isotope labeled sugars, amino acids, nucleic acids, lipid precursors, and drugs can be used to probe the dynamics of metabolic flux and interactions of drugs and cancer cells.

**Figure 7 cancers-13-01718-f007:**
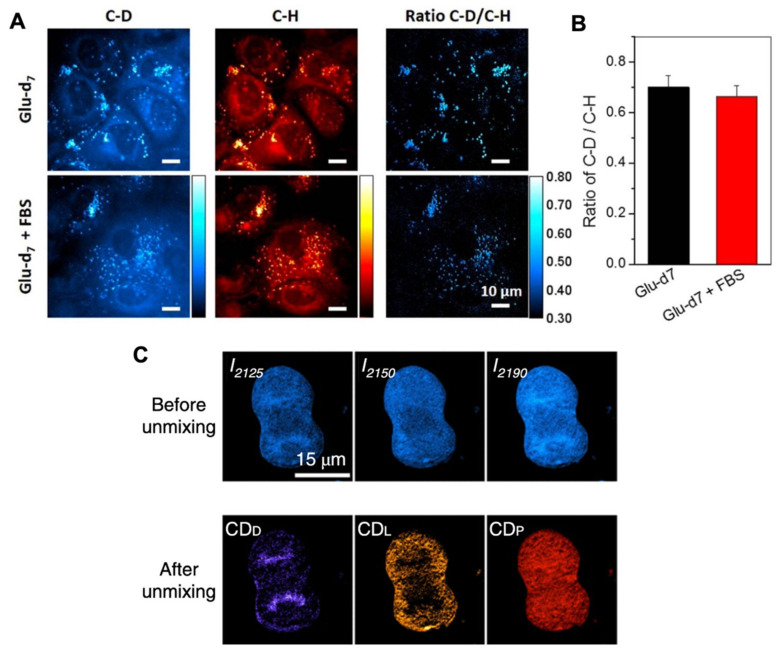
SRS imaging of cancer cells from deuterated glucose (**A**,**B**). Glucose-d_7_-incubated pancreatic cancer PANC1 cells were treated without or with 10% FBS for 3 days. SRS imaging at C–D and C–H vibration were taken and the ratio of C–D/C–H was used to analyze the level of de novo lipogenesis and increased lipogenesis. Reprinted and adapted with permission from reference [127]. (**C**) SRS images of a glucose-d7-labelled mitotic HeLa cell before and after unmixing, showing distribution of DNA, lipids, and proteins. Reprinted and adapted with permission from Reference [137].

**Figure 8 cancers-13-01718-f008:**
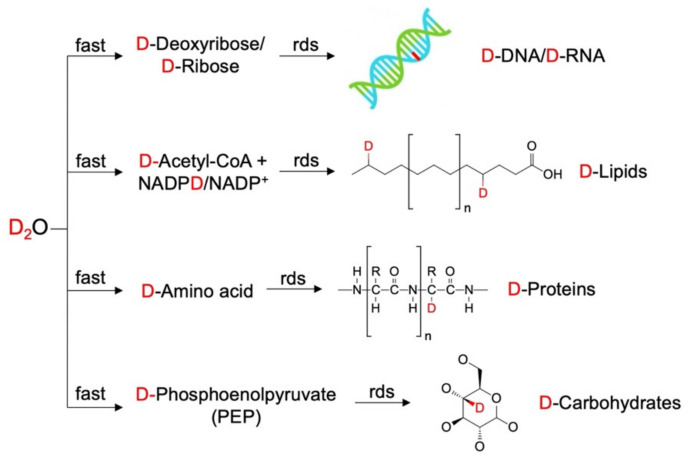
D_2_O as a unique and universal tracer for different biomolecules. ‘rds’ stands for rate-determining step.

**Table 1 cancers-13-01718-t001:** Comparison of spontaneous Raman spectroscopy and spectroscopic SRS microscopy, which are complementary to each other and can be used simultaneously.

	Spontaneous Raman	Spectroscopic SRS
**Advantages**	Relatively cost-effective and easy operation Whole spectral range with high spectral resolution	Enhanced signal Free from fluorescence and non-resonant backgroung Comparable spectrum with spontaneous Raman
**Disadvantages**	Intrinsically weak signal Fluorescence interference	Complex and expensive set-up Narrow spectral range with low spectral resolution
**Suitability**	Investigative spectral study	Targeted high-speed imaging
**Speed per spectra**	100 millisecond	20 microsecond
**Time required for a 200** **×** **200 image**	~1 hour	~1 second
**Spectral width**	Whole spectral range up to 4000 cm^‒1^	200 cm^‒1^
**Target**	Whole spectrum	Mostly CH stretching [15,16], recently also fingerprint [17]
**Spectral resolution**	~1 cm^‒1^	10 cm^‒1^

**Table 2 cancers-13-01718-t002:** Studies using Raman–SIP to probe metabolism in mammalian cells. “Spont.” refers to spontaneous Raman.

Case Studies	Spont.	CRS	Isotope	Substrate	Target	Platform
Matthäus, C. et al. (2012) [123]	√		D	d31-palmitic acid d33-oleic acid	Lipids	THP-1 monocytes
Stiebing, C. et al. (2014) [124]	√		D	d8-arachidonic acidd 31-palmitic acidd 6-cholesterol-2,2,3,4,4,6	Lipids	Human macrophages
Stiebing, C. et al. (2017) [125]	√	√	D	d31-palmitic acid	Lipids	Human macrophages
Majzner, K. et al. (2018) [126]	√		D	d8-arachidonic acid	Lipids	Endothelial cell line (HMEC-1)
Li, J. & Cheng, J.-X. (2015) [127]	√	√	D	d7-glucosed 5-glutamined 31-palmitic acid-d31	Lipids	PANC1, A549, MIA PaCa2, MCF7, LNCaP, PC3, HPDE6 and RWPE1 cell lines
García, A. et al. (2015) [128]	√	√	D	d38-cholesterol	Lipids	Y1 cell line
Weeks, T. J. et al. (2011) [129]		√	D	d2-oleic Acid-9,10	Lipids	Human monocytes
Du, J. et al. (2020) [43]	√	√	D	d7-glucose 31-palmitic acidd 35-stearic acid d33-oleic acid>	Lipids	Patient-derived melanoma cell lines
Dodo, K. et al. (2021) [130]	√		D	d-γ-Linolenic acid	γ-Linolenic acid metabolism and cytotoxicity	WI-38 cell line and VA-13 tumor cell line
Matthäus, C. et al. (2008) [131]	√		D	1,2-Distearoyl-d70-sn-glycero3-phosphocholine (DSPC-d70)	Liposomal Drug Carrier Systems	MCF-7 cell line
Van Manen, H.-J. et al. (2008) [132]	√		D	d5-phenylalanine d4-tyrosine d3-methoine	Proteins	HeLa cell line
Wei, L. et al. (2013) [133]	√	√	D	d10-leucine	Newly synthesized proteins	Live HeLa cell line Human embryonic kidney HEK293T cell line Neuron-like neuroblastoma mouse N2A cell line
Wei, L. et al. (2015) [134]	√	√	D	deuterated amino acids	Proteins	HeLa cell line
Shen, Y. et al. (2014) [135]	√	√	^13^C	^13^C-phenylalanine	Protein degradation	HeLa, HEK293T and PC12 cell lines
Miao, K. & Wei, L. (2020) [136]		√	D	d5-glutamine	Proteins	HeLa cell line
Zhang, L. *et al.* (2019) [137]	√	√	D	d12-glucose	Glucose metabolism	PC3, HeLa, MCF7, RWPE-1 and U87MG cell linesMouse model
Lee, D. et al. (2020) [138]	√	√	D	d7-glucose	Glucose metabolism; glycogen synthesis	U87 and HeLa cell lines
Hu, F. *et al.* (2015) [139]	√	√	D	3-*O* -propargyl-D-glucose	Glucose metabolism	HeLa cell line U-87 MG tumor xenograft tissue
Long, R. *et al.* (2018) [140]	√	√	D/^13^C	^13^C-3-O-propargyl-D-glucose	Glucose metabolism	U87 MG, PC-3, COS-7 and RWPE-1 cell lines
Chen, Z. *et al.* (2014) [141]	√	√	^13^C	^13^C isotopologues of EdU	DNA	HeLa cell lines
Zhang, L. & Min, W. (2017) [142]	√	√	D	d-amino acidsd31-palmitate acidd7-glucose	Lipids and proteins	MCF-7 cell lines
Shi, L. *et al.* (2018) [143]	√	√	D	D_2_O	Lipids, proteins and DNA	HeLa, COS-7, and U-87 MG cell lines Zebrafish embryos Mouse model
Hekmatara, M. *et al.* (2021) [144]	√		D	D_2_O	Lipids, proteins and DNA	MCF-7 cell line

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
