# Peer review of "Unveiling Cancer Metabolism through Spontaneous and Coherent Raman Spectroscopy and Stable Isotope Probing"

_cancers, 2021, doi:10.3390/cancers13071718_

Round 1

Reviewer 1 Report

The review by Xu and collaborators describes the recent advances in the application of spontaneous and coherent Raman Spectroscopy in the Cancer Metabolism field.  The manuscript is well-written with broad topics spanning from fundamental principles to applications and highlighting examples where this technique is surpassing mass spectrometry.

Few minor edits are suggested.

  1. For the subcellular description of metabolism (line 170) since the signal is proportional with the concentration of metabolites as the authors pointed out nicely, a whole new direction has been developed and devoted to this aspect and there are few papers that are worth adding and authors missed:
  2. https://pubmed.ncbi.nlm.nih.gov/33504762/
  3. https://pubmed.ncbi.nlm.nih.gov/30423849/
  4. https://pubmed.ncbi.nlm.nih.gov/28747639/
  5. https://pubmed.ncbi.nlm.nih.gov/31381322/
  6. https://pubmed.ncbi.nlm.nih.gov/28342983/

  1. Line 81: delete: “the incident light”, lines 82-83 please delete: “that has a molecule vibration at um and originates scattered light”.

  1. In Figure 1 for the coherent anti-Stokes Raman Scattering (CARS) it might be clearer if the authors use different colors or different/multiple arrows, to show the excitation light, as it is beautifully explained in the text.

  1. Line 102: add “ or triple bonds” after isotopes

  1. Lines 403-435: All of this is under the assumption that chemical that contain the isotope remain intact and is not transformed metabolically. Maybe just a small addition to the text should be made here. It is however more clear in section 4.1 and in Figure 8

Author Response

Response to Reviewer 1 Comments

COMMENTS FROM THE REVIEWER:

The review by Xu and collaborators describes the recent advances in the application of spontaneous and coherent Raman Spectroscopy in the Cancer Metabolism field.  The manuscript is well-written with broad topics spanning from fundamental principles to applications and highlighting examples where this technique is surpassing mass spectrometry.

Response: We thank the reviewer for the support and all the constructive comments. We have made point-to-point changes in the manuscript (highlighted) and list our responses in detail as below.

Point 1: For the subcellular description of metabolism (line 170) since the signal is proportional with the concentration of metabolites as the authors pointed out nicely, a whole new direction has been developed and devoted to this aspect and there are few papers that are worth adding and authors missed:

  1. https://pubmed.ncbi.nlm.nih.gov/33504762/
  2. https://pubmed.ncbi.nlm.nih.gov/30423849/
  3. https://pubmed.ncbi.nlm.nih.gov/28747639/
  4. https://pubmed.ncbi.nlm.nih.gov/31381322/
  5. https://pubmed.ncbi.nlm.nih.gov/28342983/

Response 1: We thank the reviewer for suggesting further readings. We have added extra discussion of “With the intensity of signal proportional to the concentration of the metabolites of interest, semi-quantitative molecular profiling of single cells and tissues is possible.”. We also added the citations that the reviewer suggested following the discussion.

Point 2: Line 81: delete: “the incident light”, lines 82-83 please delete: “that has a molecule vibration at um and originates scattered light”.

Response 2: To make the sentence clearer, we have rephrased it to “The incident laser beam with a frequency of υ0 interacts with a molecule in the sample that has a vibrational frequency of υm. This interaction originates scattered light.”

Point 3: In Figure 1 for the coherent anti-Stokes Raman Scattering (CARS) it might be clearer if the authors use different colors or different/multiple arrows, to show the excitation light, as it is beautifully explained in the text.

Response 3: We thank the reviewer for the constructive comment for making the figure more informative. We have changed the color scheme in Figure 1. Now we are using different colors for incident laser (green), Stokes (red to indicate shorter frequency), and anti-Stokes (blue to indicate longer frequency) scattering. We have also removed the curly arrows in Figure 1B, not to be confused with those in Figure 1A. Instead, we are using solid arrows for laser excitation or stimulated emission and dashed arrows for spontaneous scattering. We have also made amendments to the figure legend accordingly.

Point 4: Line 102: add “ or triple bonds” after isotopes

Response 4: We have added the description of “…and can involve bands contributed by stable isotopes or triple bonds”.

Point 5: Lines 403-435: All of this is under the assumption that chemical that contain the isotope remain intact and is not transformed metabolically. Maybe just a small addition to the text should be made here. It is however more clear in section 4.1 and in Figure 8

Response 5: We thank the reviewer for raising this point. We have included an additional description in the Introduction part of Section 4: “Among those stable isotopes, D (2H) and 13C are mostly commonly used due to the stronger C–H bond strength. Heteroatomic X–H bonds such as O–H, N–H, and S–H can be exchanged from nonenzymatic reactions while C–H bond formation depends solely on enzyme-catalyzed chemical reactions happened within the metabolic pathways. That provides unique advantages of D (2H) and 13C for probing biological metabolism.”

Reviewer 2 Report

The paper entitled ‘Spontaneous and Coherent Raman Spectroscopy to Study Cancer Metabolism’ by
Jiabao Xu et al. shows a short review on cancer metabolism studied by Raman-based techniques. The
authors focused on both spontaneous and coherent Raman spectroscopy highlighting isotope probing
applications. Since these advanced methods of Raman spectroscopy can be potentially used for clinical
applications, the manuscript can be of interest to the journal’s readers. However, the manuscript
should undergo a minor revision before publication. The main issues are listed below.
1. Since a large part of the manuscript is devoted to isotope probing, I am wondering if the title
of the manuscript can be extended to cover this type of study. Additionally, I suggest
including isotope probing in manuscript keywords.
2. Fig. 1B – If the authors use a straight line for stimulated transitions and a waved line for
spontaneous ones, why do you use a straight line for the v(aS) transition in CARS?
3. Page 3, line 95 – ‘containing hundreds, if not thousands of Raman bands’ – the sentence is
confusing. Raman spectra do not contain hundreds (of separated) bands. Please correct the
sentence.
4. Page 3, line 97 – ‘Raman spectrum of a single glioblastoma cell with major biological
macromolecules labelled’ – it is colloquialism … you do not observe molecules in spectra but
bands related to vibrations in molecules. Please correct the sentence.
5. Page 3, line 113 – ‘does not hinder sample signals in Raman spectroscopy’ – it is not always
true. Please say that Raman bands of water are relatively weak and can be easily subtracted
during the preprocessing procedure.
6. Fig. 2 – in the range of 1520-1700 cm-1 you observe Amide I band, not Amide III band.
Additionally, if you use names of biomolecules, you should label the high wavenumber region
as ‘proteins, lipids’ instead of ‘C-H band’.
7. Page 4, line 126 – ‘It unbiasedly probes all macromolecules and collectively displays them in
one spectrum’ – It is colloquialism as well … Raman spectroscopy does not display molecules
in a spectrum! Please rewrite the sentence.
8. Page 5, line 141 – ‘generating either a Stokes signal νS or an anti-Stokes signal νaS.’ – the
pump beam generates both Stokes and anti-Stokes signals simultaneously! It is not true that
it generates either Stokes or anti-Stokes! You usually acquire either Stokes or anti-Stokes but
they are both generated by a molecule.
9. Table 1 – why do not you compare all three mentioned methods, i.e. spontaneous Raman,
SRS, and CARS? For speed per spectra, it is more suitable to use periods like you used in the
text (seconds to minutes and microseconds). For spectral width, please use up to 4000 cm-1
for RS. For target, please clarify mostly CH stretchings rather than just CH (CH bendings are
present in the fingerprint region).
10. Page 6, line 197 – ‘As the lipid CH vibrations are’ – CH stretching vibrations as CH bendings
are not in the 2800-3200 cm-1 region. The same in Page 8, line 291.
11. Page 7, line 254 – ‘Raman peaks at 1655 and 1245 cm–1.’ – Please provide the spectral range
for both Amide bands as not all proteins show Amide bands exactly at 1655 and 1245 cm–1.
12. Fig. 4 – phenylalanine, tyrosine, and Amide I band should show the same distribution in a cell
as all these bands come from proteins. Why the Phe signal is limited mostly to the cell
nucleus?
13. Fig. 5. – figure caption – you used ‘5 mm’ instead of ‘5 µM’
14. Page 10, line 354 – missing ‘l’ in ‘et al.’
15. Page 13, line 464 – in the equation you have Area(C-D)/Area(C-H), whereas the text states
‘quantify D incorporation is to calculate the percentage of the integrated C–D band area to
the sum of the areas of the C–D and C–H bands’. Please correct the equation.
16. Page 17, line 593 – you repeated ‘and’ twice

Author Response

March 23, 2021

Re: Revision of “Spontaneous and Coherent Raman Spectroscopy to Study Cancer Metabolism”

Dear Editorial Board,

R

Response to Reviewer 2 Comments

COMMENTS FROM THE REVIEWER:

The paper entitled ‘Spontaneous and Coherent Raman Spectroscopy to Study Cancer Metabolism’ by
Jiabao Xu et al. shows a short review on cancer metabolism studied by Raman-based techniques. The
authors focused on both spontaneous and coherent Raman spectroscopy highlighting isotope probing
applications. Since these advanced methods of Raman spectroscopy can be potentially used for clinical
applications, the manuscript can be of interest to the journal’s readers. However, the manuscript
should undergo a minor revision before publication. The main issues are listed below.

Response: We are grateful for the reviewer’s comments and support. We have made point-by-point changes in the manuscript (highlighted) and list our responses in detail as below.

Point 1: Since a large part of the manuscript is devoted to isotope probing, I am wondering if the title of the manuscript can be extended to cover this type of study. Additionally, I suggest including isotope probing in manuscript keywords.

Response 1: We have extended the title and key words to include stable isotope probing. The title has been changed to “Unveiling Cancer Metabolism by Spontaneous and Coherent Raman Spectroscopy and Stable Isotope Probing”.

Point 2: Fig. 1B – If the authors use a straight line for stimulated transitions and a waved line for spontaneous ones, why do you use a straight line for the v(aS) transition in CARS?

Response 2: We thank the reviewer for the raising the point. Straight lines in CARS represent stimulated emission instead of spontaneous emission. Using four straight lines in CARS scheme diagram is also more common–please refer to the two review papers that use similar illustrations:

  • J. Tipping, W.; Lee, M.; Serrels, A.; G. Brunton, V.; N. Hulme, A. Stimulated Raman Scattering Microscopy: An Emerging Tool for Drug Discovery. Chemical Society Reviews 2016, 45 (8), 2075–2089. https://doi.org/10.1039/C5CS00693G.
  • (1)
  • (1) Zhang, C.; Zhang, D.; Cheng, J.-X. Coherent Raman Scattering Microscopy in Biology and Medicine. Annual Review of Biomedical Engineering 2015, 17 (1), 415–445. https://doi.org/10.1146/annurev-bioeng-071114-040554.

However, to make Figure 1 more intuitive and informative, we have changed the color scheme as well as the line shapes. Now we are using different colors for incident laser (green), Stokes (red to indicate shorter frequency), and anti-Stokes (blue to indicate longer frequency) scattering. We have also removed the curly arrows in Figure 1B, not to be confused with those in Figure 1A. Instead, we are using solid arrows for laser excitation or stimulated emission and dashed arrows for spontaneous scattering. We have also made amendments to the figure legend accordingly.

Point 3: Page 3, line 95 – ‘containing hundreds, if not thousands of Raman bands’ – the sentence is confusing. Raman spectra do not contain hundreds (of separated) bands. Please correct the sentence.

Response 3: To make the sentence clearer, we have deleted “hundreds, if not thousands of”.

Point 4: Page 3, line 97 – ‘Raman spectrum of a single glioblastoma cell with major biological macromolecules labelled’ – it is colloquialism … you do not observe molecules in spectra but bands related to vibrations in molecules. Please correct the sentence.

Response 4: We have changed the sentence to “Figure 2 illustrates a Raman spectrum of a single glioblastoma cell with Raman bands labeled with assignments of major biological macromolecules.”

Point 5: Page 3, line 113 – ‘does not hinder sample signals in Raman spectroscopy’ – it is not always true. Please say that Raman bands of water are relatively weak and can be easily subtracted during the preprocessing procedure.

Response 5: We have changed the sentence to “A water molecule has very low polarizability, thus minimally hinder sample signals in Raman spectroscopy and can be easily subtracted during preprocessing procedures.”

Point 6: Fig. 2 – in the range of 1520-1700 cm-1 you observe Amide I band, not Amide III band. Additionally, if you use names of biomolecules, you should label the high wavenumber region as ‘proteins, lipids’ instead of ‘C-H band’.

Response 6: We have changed the labeling in Figure 2 from vibrational modes to biomolecules. “Amide” and “C-H” bands have been changed to proteins, lipids and nucleic acids.

Point 7: Page 4, line 126 – ‘It unbiasedly probes all macromolecules and collectively displays them in one spectrum’ – It is colloquialism as well … Raman spectroscopy does not display molecules in a spectrum! Please rewrite the sentence.

Response 7: We have changed the sentence to “It unbiasedly probes all macromolecules and collectively displays the vibrational modes of them in one spectrum.”

Point 8: Page 5, line 141 – ‘generating either a Stokes signal νS or an anti-Stokes signal νaS.’ – the pump beam generates both Stokes and anti-Stokes signals simultaneously! It is not true that it generates either Stokes or anti-Stokes! You usually acquire either Stokes or anti-Stokes but they are both generated by a molecule.

Response 8: We have changed the sentence to “During spontaneous Raman scattering, the pump beam with a frequency of ν0 is incident upon the sample generating a Stokes signal νS and an anti-Stokes signal νaS.”

Point 9: Table 1 – why do not you compare all three mentioned methods, i.e. spontaneous Raman,
SRS, and CARS? For speed per spectra, it is more suitable to use periods like you used in the text (seconds to minutes and microseconds). For spectral width, please use up to 4000 cm-1 for RS. For target, please clarify mostly CH stretching rather than just CH (CH bending are present in the fingerprint region).

Response 9: In Table 1, we have changed the units of speed and time to “hour”, “second”, “millisecond”, and “microsecond”. We have changed the spectral width to “up to 4000 cm–1”. We have changed the target description to “Mostly CH stretching”.

We thank the reviewer for commenting that three methods can all be compared together. Our argument is that this table is to show the complementary features of spontaneous Raman and SRS, and that they can be used simultaneously. As of CARS, the strong non-resonant background can distort the spectra and make the comparison with spontaneous Raman difficult. That is the reason that we include only spontaneous Raman and SRS in the table. We have added an additional sentence to explain the logics: “SRS would be a complementary tool to spontaneous Raman scattering and simultaneously using the two can be beneficial (Table 1)”.

Point 10: Page 6, line 197 – ‘As the lipid CH vibrations are’ – CH stretching vibrations as CH bendings
are not in the 2800-3200 cm-1 region. The same in Page 8, line 291.

Response 10: We have changed line 197 to “As the lipidic CH stretching vibrations are the strongest among all Raman vibrations…” and line 291 to “…compared to in the high-frequency CH stretching region,…”.

Point 11: Page 7, line 254 – ‘Raman peaks at 1655 and 1245 cm–1.’ – Please provide the spectral range
for both Amide bands as not all proteins show Amide bands exactly at 1655 and 1245 cm–1.

Response 11: We have changed the description from one wavenumber to a spectral range for more accuracy: “ The study of proteins can be conducted by investigating Raman bands of Amide vibrational modes, including Amide I that ranges from 1600–1670 cm–1, Amide II that ranges from 1480–1580 cm–1, and Amide III that ranges from 1230–1300 cm–1.”

Point 12: Fig. 4 – phenylalanine, tyrosine, and Amide I band should show the same distribution in a cell
as all these bands come from proteins. Why the Phe signal is limited mostly to the cell
nucleus?

Response 12: We thank the reviewer for pointing this out. We agree that phenylalanine signal should have similar distribution as the proteins, and the 1003 cm–1 map signal in Figure 4 could be mixed with nucleic acid content without undertaking a unmixing procedure. To avoid confusion, we have removed the 1003 cm–1 map in Figure 4.

Point 13: Fig. 5. – figure caption – you used ‘5 mm’ instead of ‘5 µM’

Response 13: We have corrected it into “5 µM”.

Point 14: Page 10, line 354 – missing ‘l’ in ‘et al.’

Response 14: We have made correction to “et al.”.

Point 15: Page 13, line 464 – in the equation you have Area(C-D)/Area(C-H), whereas the text states
‘quantify D incorporation is to calculate the percentage of the integrated C–D band area to
the sum of the areas of the C–D and C–H bands’. Please correct the equation.

Response 15: We thank the reviewer for spotting the mistake in the equation. We have corrected the equation according to the text (Dincorp = AreaC-D /  (AreaC-D + AreaC-H) ).

Point 16: Page 17, line 593 – you repeated ‘and’ twice

Response 16: We have removed the extra “and”.